# Novel Insights into Milk Coffee Products: Component Interactions, Innovative Processing, and Healthier Product Features

**DOI:** 10.3390/foods14234043

**Published:** 2025-11-25

**Authors:** Yi Li, Dan Zhao, Weili Li, Xiaoyan Yu, Xueting Zhang, Jindou Shi, Hong Li, Yanan Shi

**Affiliations:** 1Yunnan College of Modern Coffee Industry, Yunnan Agricultural University, Kunming 650201, China; ly1186815404@icloud.com (Y.L.); 13428848324@163.com (W.L.); 2College of Food Science &Technology, Yunnan Agricultural University, Kunming 650201, China; minyanzhiliu@163.com (D.Z.); yxy001023@126.com (X.Y.); meinixueting@126.com (X.Z.); sh488709976@163.com (J.S.)

**Keywords:** milk coffee, molecular interactions, innovative processing, innovative products

## Abstract

Milk coffee is a composite beverage in which interactions among dairy proteins, lipids, and coffee polyphenols govern flavor, texture, and stability. This review synthesizes recent research to guide formulation and processing, covering conventional Ultra-high temperature sterilization (UHT) and innovative routes including blending-after-sterilization (BAS), high-pressure homogenization (HPH), ultrasound/pulsed electric field (PEF)/cold plasma (CP), microencapsulation, and plant-based matrices. Key findings show that non-covalent protein–polyphenol complexes and interfacial partitioning at fat-globule membranes control volatile retention, astringency, droplet structure, and phenolic bioaccessibility; appropriate fat levels and HPH refine microstructure; BAS better preserves aroma; and matrix or decaffeination choices modulate antioxidant capacity. Guided by these insights, we propose a concise “process–activity–stability” framework linking parameters to functionality and shelf life to accelerate the development of high-quality, nutritious, enjoyable, and more sustainable milk coffee products.

## 1. Introduction

Milk coffee, a reconstituted dairy drink, is prepared by mixing milk and other dairy products with coffee in varying proportions. It is typically manufactured via acidity adjustment, sweetening, emulsification, sterilization, and aseptic filling, yielding a beverage with distinctive color and flavor as well as added nutritional and functional attributes. Recent market data underscore its momentum: in 2022, the US coffee market reached $76.7 billion, with the milk-coffee sector accounting for $57.5 billion; China reached $62.6 billion with 24.7% year-on-year growth [1]. Together, these trends highlight strong prospects for processing innovation and product development.

However, comprehensive reviews dedicated specifically to milk coffee remain relatively scarce. Existing coffee research mainly covers industry-chain analyses [2], valorization of co-products [3], Arabica flavor and fermentation chemistry [4], and health-oriented work on polyphenols and caffeine (Figure 1; [5,6]). Accordingly, and in line with recent advances directly relevant to milk-coffee systems, this review (i) integrates component-level mechanisms—protein–polyphenol and lipid–polyphenol interactions, supported by thermodynamic and spectroscopic evidence—and their consequences for aroma retention, astringency, emulsion structure, and phenolic bioaccessibility; (ii) synthesizes processing technologies (BAS, HPH, ultrasound/PEF/cold plasma) in comprehensive comparison with UHT, providing concise quantitative contrasts where available; and (iii) incorporates low-temperature cold-brew-type pathways to elucidate flavor and antioxidant retention in dairy matrices. On this basis, we first summarize common milk-coffee manufacturing processes and key control points and then propose innovation-oriented technologies for production.

The contribution of this study is significant for the advancement of the dairy coffee industry. Despite rapid market growth and abundant primary research, integrated syntheses on milk coffee that are truly processing-relevant remain relatively scarce. By mapping protein–polyphenol and lipid–polyphenol mechanisms to measurable product attributes (aroma retention, astringency, emulsion stability, and phenolic bioaccessibility), standardizing terminology and reconciling divergent conclusions (bioaccessibility = in vitro release and GI stability; bioavailability = in vivo absorption and kinetics), and critically evaluating processing routes through concise quantitative contrasts—benchmarking BAS, HPH, and PEF/CP against UHT, and incorporating low-temperature (cold-brew) pathways and MD/spectroscopic readouts—this work provides actionable guidance for formulation, scale-up, and sustainability assessment (e.g., process analytical technology, digital twins, and LCA/TEA) to meet evolving consumer needs and support market expansion, ultimately leading to a more diverse and dynamic milk-coffee landscape.

## 2. Overview of Milk Coffee Research

According to statistics from Citexs on 16 February 2024, the number of academic articles related to milk coffee has been on the rise, with 544 articles published from 2014 to 2024, averaging 55 publications per year. The peak was observed in 2020 with 98 articles, marking a rapid growth rate of 96%. This surge indicates a significant and growing interest in the nutritional, functional, and environmental aspects of milk coffee.

The United States leads in milk coffee research with 64 publications, followed by China with 58 and India with 49, as shown in Figure 2A. The multidimensional research landscape of milk coffee focuses on the quality of raw materials, including caffeine, polyphenols, chlorogenic acid, milk proteins, and antioxidants, which are pivotal for product development due to their influence on stability, bioaccessibility, and flavor profile, illustrated in Figure 2B.

The multidimensional research landscape of milk coffee. Milk coffee, a combination of milk and coffee, is a globally popular beverage celebrated for its rich nutritional and sensory attributes. Research in this field highlights the critical role of raw material quality, focusing on components such as caffeine, polyphenols, chlorogenic acid, milk proteins, and antioxidants. These elements influence the stability, bioaccessibility, and flavor profile of milk coffee, making them essential for product development [5,6,7,8,9]. In addition to nutritional aspects, milk coffee research addresses evolving consumer demands for sustainable and plant-based alternatives. Studies, such as Rashidinejad et al. [5] have demonstrated the health benefits of milk coffee, including improved digestion and reduced inflammation due to its bioactive compounds. These findings, coupled with milk coffee’s favorable sensory profile, underscore its appeal to modern consumers. The bioactive compounds in milk coffee may influence digestion and inflammatory responses through multiple mechanisms. Chlorogenic acid possesses antioxidant and anti-inflammatory properties, potentially reducing the release of inflammatory factors by inhibiting the NF-κB pathway while modulating the balance of gut microbiota. The proteins in milk can buffer stomach acid, mitigating coffee’s irritation to the gastric mucosa, and the milk fat globule membrane may enhance intestinal barrier function. Additionally, caffeine at low doses can promote intestinal motility, while calcium ions in milk may help regulate intestinal contractions.

Milk coffee research currently focuses on key areas: (i) Advanced Extraction Techniques. Recent research has delved into novel extraction methods for coffee, such as cold brew and nitro coffee, with potential applications in milk coffee. Cold brew extraction, in particular, provides a valuable model for studying the retention of coffee polyphenols and the modulation of sensory attributes under low-temperature conditions, offering insights into how flavor balance and antioxidant stability are maintained [10,11]. Investigations by Yang et al. [12] and Chen et al. [13] have scrutinized the impact of these techniques on flavor profiles and nutritional content. (ii) Microencapsulation of Coffee Compounds. Microencapsulation technology effectively protects coffee compounds by encapsulating them within tiny capsules, shielding them from external environmental factors such as light, oxygen, moisture, and temperature. This prevents the oxidation and deterioration of flavor components or slows down the oxidation process. In beverages like milk coffee, microencapsulation can significantly prolong the release time of flavor substances, thereby maintaining the persistence of the flavor. Carmo et al.’s studies in 2022 have investigated the microencapsulation of coffee compounds to enhance flavor stability and nutritional content in milk-based coffees, presenting a promising technological innovation within the industry [14]. Sensory analysis showed that different microencapsulation methods had different effects on coffee flavor retention. The flavor acceptance of the freeze-drying (FD) process sample was significantly higher than that of the spray drying (SD) process. Among them, the flavor score of the green coffee extract treated by freeze-drying (IN-FD) with inulin (IN) as the wall material was significantly higher than that of the SD treatment sample (PD-SD, IN-SD) after being added to the dairy beverage, and there was no significant difference between the freeze-drying (FD) process sample and the control group without addition, while the flavor score of the SD treatment sample was significantly lower than that of the control group. (iii) Use of Enzymes in Coffee Processing. De melo pereira et al. [15] have focused on utilizing enzymes to improve the extraction process during coffee production, with potential applications to enhance flavor profile while reducing bitterness in milk coffees. (iv) Sustainable Coffee Production. Studies conducted by Tongcumpou et al. [16] and Lee et al. [17] have centered around sustainable practices within coffee production that can be integrated into milk coffee processing to promote environmental sustainability. (v) Health Benefits of Milk Coffee. Research by Machado et al. [18] has looked into the health benefits of coffee, including its antioxidant and anti-inflammatory properties. Extending this research to milk coffee could provide insights into the health implications of different processing techniques. (vi) Consumer Preferences and Sensory Analysis. Studies by Sergio [19] and Cai et al. [20] have used sensory analysis to understand consumer preferences for different coffee products. This information could be valuable in developing new milk coffee products that cater to consumer tastes.

Current research on milk coffee centers on four core dimensions: technological innovation, industrial sustainability, health value, and market demand. It has laid a solid foundation for the development of the industry. Based on the existing research foundation and industrial development trends, future research on milk coffee should move toward technological integration. For instance, by combining microencapsulation technology with techniques such as molecular gastronomy and 3D printing, energy microcapsule technology can be utilized to achieve the precise release of flavor substances, thereby promoting the establishment of precise flavor regulation technology. Meanwhile, research should focus on the high-value utilization of by-products like coffee husks, whey and coffee grounds so as to construct a circular and sustainable development system.

## 3. Component Interactions in Milk Coffee

The development of milk coffee with an emphasis on taste and functional nutrition is a dynamic field that requires a deep understanding of consumer preferences, food science, and nutrition. Dairy products, such as milk, are added to coffee in varying proportions. Common additions include whole milk, semi-skimmed or skimmed milk, light or heavy cream, condensed milk, and plant-based alternatives, all of which are popular.

Consumers regard dairy products as an ideal source of nutrients including protein, fat, and lactose. The health benefits of coffee stem from its functional polyphenols, which include chlorogenic acid, caffeic acid, ferulic acid, and p-coumaric acid. These compounds exhibit various health-promoting benefits, such as antioxidants, antibacterial properties, immune enhancement, liver protection, and hypoglycemic effects [21], as illustrated in Figure 1. Polyphenols and proteins are often found together in foods. Therefore, studies on protein-polyphenol, milk fat-polyphenol, protein-flavor, and milkfat-flavor interactions offer prospects for the development of functional dairy drinks with novel nutritional and flavor components.

### 3.1. Interaction Between Milk Proteins and Polyphenols

Protein-polyphenol interactions affect the structure, molecular weight, stability, and functional properties of proteins and polyphenols [22]. Polyphenols interact with proteins in the food matrix through non-covalent and covalent interactions [23]. These interactions, which include hydrophobic and hydrogen bonding, can alter the solubility of proteins, affecting the creaminess and shelf life of the product. Understanding these dynamics is essential to refining the flavor, texture and health benefits of milk coffee.

Of these, the primary driving forces for protein-polyphenol complex formation are hydrophobic interactions and hydrogen bonding [24]. Chlorogenic acid, the main polyphenol in coffee, is classified into three categories: caffeoylquinic acid (5-o-caffeoylquinic acid is the most abundant), ferroylquinic acid, and dicaffeoylquinic acid [25]. Chlorogenic acid interacts with α-lactalbumin and α-casein by hydrogen bonding and van der Waals forces, whereas it interacts with β-lactoglobulin (β-LG), β-casein, and κ-casein through hydrophobic binding [26], as shown in Figure 1. Recent thermodynamic and spectroscopic assessments support these binding modes: fluorescence quenching and isothermal titration calorimetry typically indicate spontaneous, enthalpy–entropy–coupled complexation (negative ΔG with hydrophobic and hydrogen-bond contributions), while circular dichroism and FTIR reveal subtle secondary-structure rearrangements upon complex formation [27,28,29].

Protein-polyphenol interactions mainly affect the hydrophobicity, solubility, thermal stability, and hydrolytic stability of proteins. Hydrophobic chlorogenic acid can induce partial unfolding of protein structures and reduce the hydrophobicity of protein surfaces [30]. The hydrogen bonding and van der Waals interactions played major roles in the β-LG binding process with chlorogenic acid. This interaction causes the secondary structure of β-LG to change from an α-helical structure to a β-structure, disrupting the hydrophobic structure of the protein [31]. Their interaction significantly improves the solubility and antioxidant properties of β-LG and the thermal stability of lactoglobulin [32]. Furthermore, in the protein-coffee polyphenol complex, the protein can act as a physical separation barrier, protecting the polyphenols from oxidation and enzymatic hydrolysis during gastrointestinal digestion [33,34]. The complexes formed through interactions between polyphenols and proteins can diminish the degradation of polyphenols in the intestinal tract and elevate the concentration of intact polyphenols. When polyphenols occupy the hydrophobic regions of proteins, they can impede digestive enzymes from accessing the active sites within these regions, thereby reducing enzymatic hydrolysis of polyphenols and indirectly safeguarding them against enzymatic breakdown. Additionally, the complex structure formed by proteins and polyphenols serves as a physical barrier, minimizing direct contact between the active components and the external oxidative environment, thus protecting polyphenols from oxidative degradation. These mechanistic readouts help explain why protein–polyphenol complexation can protect coffee polyphenols during processing and digestion and sustain antioxidant performance in milk–coffee matrices [28,35].

### 3.2. Interaction Between Milk Fat and Polyphenols

Kamiloglu et al. [8] studied the hydrophobic interactions between coffee polyphenols and milk fat. Lipids can “trap” polyphenols and form polyphenol micelles, improving the colloidal stability of polyphenol-lipid mixed micelles in the gastrointestinal tract [36]. Alongi et al. [37] investigated the effects of different fat concentrations and HPH on the bioaccessibility of chlorogenic acid in milk coffee. They discovered that samples with a higher fat content had the lowest zeta potential and the smallest particle size distribution during in vitro digestion. These data imply that a higher fat content may enhance the stability of chyme formation. Milk fat reduces the susceptibility to the degradation of phenolic substances including caffeic acid and chlorogenic acid through high-pressure homogeneous micellization, while the interaction between milk fat and polyphenols is exploited in the production of nanocapsules, serving as an effective carrier for polyphenols in the gastrointestinal tract. As illustrated in Figure 3. The increase in milk fat increases the level of surfactants in coffee, preventing the precipitation of polyphenols and thereby increasing their bioaccessibility. Consequently, the interaction between milk fat and polyphenols helps protect the polyphenols from degradation and improves their absorption rate in the body [21].

Complementary spectroscopic and molecular-simulation evidence indicates that coffee polyphenols preferentially accumulate at lipid and milk fat globule membrane interfaces, stabilized by non-covalent contacts with polar lipids and interfacial proteins; this interfacial localization rationalizes the observed gains in polyphenol stability and bioaccessibility in higher-fat systems [38].

### 3.3. Sensory and Flavor Implications of Component Interactions

Effect on milk coffee sensory. Future research is likely to focus on using molecular sensory science to better understand how these interactions affect the sensory properties of milk coffee, such as taste, aroma and mouthfeel. Vision plays an important role in the eating experience, and the appearance of food, particularly its color, can influence flavor perception and recognition [39]. Increasing the color intensity of foods and beverages results in higher flavor-intensity ratings [40]. Thus, manipulating the visual appearance of food can help to encourage healthy eating behaviors. Light brown coffee is more creamy than dark brown coffee, while the color of milk coffee does not affect perceived sweetness or preference. The pattern or design of the surface of milk coffee affects the consumer’s neural activity in the brain, increasing their perception of sweetness [41], as shown in Figure 4. Currently, the coffee market offers a diverse array of trendy beverages, including visually striking Dirty Coffee and indulgent dessert-inspired coffee options, both of which have gained immense popularity among young consumers. Among these, Dirty Coffee stands out as a uniquely dynamic drink that combines the bold intensity of espresso with the smooth creaminess of cold milk, creating a one-of-a-kind and highly enjoyable sensory experience.

Effect of milk addition on coffee astringency. Astringency is often considered an important flavor characteristic of coffee; however, excessive astringency can lead to unpleasant sensations and thus diminish consumer acceptability. As a result, coffee drinks are often suggested in combination with cream (e.g., milk) and sweeteners (e.g., sugar). Increasing the proportion of whole milk in coffee drinks contributes to the formation of a more complete and stable lubricating film with a lower coefficient of friction, resulting in a sweet taste [42]. Therefore, it is recommended to increase the proportion of whole milk in coffee drinks to 30% and above.

Effect of milk addition on coffee flavor. The addition of milk has a greater impact on medium- and dark-roasted coffees. Flavor release is affected by the addition of milk fat, and its extent depends on the lipophilicity of flavor compounds. Song et al. [43] analyzed the influence of milk components on volatiles in the top air of coffee by HS-SPME-GC-MS. Non-fat milk solids reduced the release of 6 compounds (N-ethylpyridine, 3,3-dimethacrylic acid, p-methoxyphenthiophenol, β-damalenone, and 2-acetyl-1-methylpyrrole) while facilitating the release of 12 compounds (2,4-Dimethylfuran, 2-ethyl-5-methylpyrazine, isobutyraldehyde, hexanal, Pyridine, 2-methylpyridine, 3-methylpyridine, 2,5-dimethylpyridine, 3-Hexanone, 4,5-dimethylthiazole, Octanoic acid, Decanoic acid), which is the result of the combined effect of the binding and repulsion of whey protein and casein in non-fat milk solids, and the salting out of lactose and salt. However, milk fat retains hydrophobic volatiles; the more hydrophobic a volatile compound is, the slower its release rate. This mechanism inhibits the release of 62 volatile compounds in coffee (Figure 4) [2]. The effect of milk on medium- and dark-roasted coffee essentially lies in optimizing the rhythm of flavor release, balancing taste, and reconstructing mouthfeel through interactions between components. Ultimately, it transforms the strong and bitter characteristics of medium- and dark-roasted coffee into a complex flavor profile characterized by richness, smoothness, and a balance between sweetness and bitterness, which is more in line with the sensory preferences of the general public for mild-style coffee. Similarly, Bücking and Steinhart [44] suggested that aromatic compounds in coffee, particularly 2-furanoyl mercaptan, are influenced by van der Waals forces and exhibit a dipole interaction with fat. This interaction eventually reduces the release of volatile flavor compounds in milk coffee. Milk coffee with (0.3%milk fat) low-fat milk contained more aromatic compounds than milk coffee with (10%milk fat) high-fat milk did. Akiyama et al. [45] found that adding 0.5% milk fat to milk coffee improves its flavor. Therefore, adding milk with a lower fat content (approximately 0.5%) to milk coffee yields a better flavor.

### 3.4. Antioxidant Capacity and Bioaccessibility Related to Component Interactions

Unraveling the effects of milk on coffee polyphenols: (i) Recent studies have indicated that the incorporation of milk into coffee can affect its nutritional, functional, and sensory attributes [5]. The outcome is influenced by various factors such as the milk-to-coffee ratio, temperature, the preparation method of the coffee, type of milk used, and the analytical endpoint used to assess antioxidant properties. Dose–response studies on bioaccessibility (in vitro release/stability) and bioavailability (in vivo absorption/kinetics) are also becoming more prevalent. (ii) Interactions between milk fat and phenolic compounds (catechins) in other types of beverages have been reported [46]. These findings offer mechanistic analogs for milk–coffee systems. A 2022 study further reported that binding of coffee polyphenols to milk-fat-globule membranes was associated with neuroprotective outcomes, including inhibition of brain aging and improvement of memory [9]. Against this background, the phenolic content and antioxidant activity of milk coffee are greatly influenced by the milk substrate and processing techniques used, which are crucial in the processing and development of innovative milk coffee products.

Across studies summarized in Appendix A, the direction and magnitude of changes in antioxidant activity following milk addition depend on the assay used and the matrix: decreases are typically reported when chemical assays quantify free phenolics or radical-scavenging capacity in extracts [47,48]; several in vitro digestion protocols show no material change [7,21,49]; and increases are observed when protein–phenol complexation and fat-assisted micellar solubilization protect or better dissolve phenolics, including in decaffeinated matrices [50,51,52,53,54]. For example, Soares et al. [50] demonstrated that adding 50% milk to decaffeinated coffee improved antioxidant benefits. Moreover, caffeine was found to have a pro-oxidant effect, which might cause the oxidation of specific phenolic acids during roasting. Conventional coffee exhibited lower concentrations of phenolic acids, whereas decaffeinated coffee and decaffeinated milk coffee had the highest phenolic acid levels among all samples. Both decaffeinated coffee and decaffeinated milk coffee displayed increased antioxidant activity following in vitro digestion. Across studies summarized in Appendix A, reported effects of milk on polyphenol bioaccessibility diverge: HPLC-tracked analyses often find lower chlorogenic-acid availability when milk is added [47,55]; co-digestion with milk plus sugars or sweeteners increases total flavonoid availability [8]; skim-milk systems dominated by protein effects can reduce availability [6]; whereas micellization of chlorogenic acids and the use of decaffeinated coffee are associated with improved bioaccessibility [34,37,50]. Taken together, these mixed outcomes underscore the need to clarify mechanisms: here, bioaccessibility refers to release from the food matrix and gastrointestinal (GI) stability, whereas bioavailability additionally depends on intestinal absorption efficiency, metabolism, and host factors [56,57]. Milk addition alters the matrix composition and can modulate absorption efficiency, as well as release and transformation, during digestion.

Kamiloglu et al. [8] studied the effects of milk, sugar, and sweetener addition on the bioaccessibility of terebinth coffee polyphenols. Terebinth coffee, whole milk, and sugar in a volume ratio of 9.6:24.3:1 demonstrated the highest bioaccessibility of flavonoids and non-flavonoids. The inclusion of skim milk had no significant effect on the results. Co-digestion of sugar or sweeteners with flavonoid-rich foods can improve the overall digestion and absorption of flavonoids, increasing their content and bioaccessibility during intestinal digestion [58]. Sugars or sweeteners can increase the solubility of flavonoids in gastrointestinal fluids by enhancing the osmotic pressure of the digestive environment or forming hydrogen bonds with flavonoids, thereby promoting their release from food matrices. Also, sugars or sweeteners can reduce pH fluctuations by stabilizing the digestive environment, providing a more stable chemical environment for flavonoids.

Alongi et al. [37] discovered that higher fat concentrations were found to promote the formation of small micelles during digestion. When the fat content of milk was increased (0.1% to 3.6%) and treated under the appropriate homogeneous conditions, an increase in phenolic bioaccessibility was observed. Quan et al. [6] found that coffee blended with whole milk (40% coffee, 56% whole milk powder, 4% water) had higher phenolic bioaccessibility than pure coffee or coffee with skim milk. Whole milk enhances phenolic bioaccessibility through protein-phenol complexes and lipid interactions, which protect phenolics during digestion [49].

In order to obtain milk coffee with better sensory properties and nutritional functions, there are several key points: (i) the key to maintaining the antioxidant benefits and bioaccessibility of coffee in milk coffee beverages lies in the proportion of milk added. A balanced approach with a moderate amount of milk, not exceeding 20% of the coffee volume, can help preserve the healthful properties of coffee without significantly affecting the bioaccessibility of its antioxidants. (ii) when milk with a higher fat concentration, such as whole milk, is added to coffee, it can enhance the bioaccessibility of the coffee’s compounds. This increased bioaccessibility suggests that the fat in milk may aid in the absorption and utilization of coffee’s bioactive substances, potentially leading to greater health benefits. (iii) The choice of decaffeinated coffee and the addition of milk to coffee may confer greater antioxidant activity to coffee drinks. In particular, decaffeination can remove pro-oxidant caffeine and elevate measurable phenolic acids in some matrices, contributing to higher post-digestion antioxidant activity [50]. Because much of the current evidence is derived from standardized in vitro digestion and spectroscopic/thermodynamic readouts that simplify real matrices and do not capture epithelial transport, active efflux, mucus dynamics, or phase-II metabolism, these practical levers should be optimized with caution. Likewise, benefits demonstrated under controlled in vitro conditions require corroboration in matched formulations to establish whether they translate to higher systemic exposure. Accordingly, direct in vivo or ex vivo transport confirmation is still needed to determine the extent to which gains in bioaccessibility are accompanied by genuine increases in bioavailability. Specific thresholds should be optimized case-by-case rather than assumed universally.

## 4. Processing Technologies in Milk Coffee Systems

### 4.1. Standard Manufacturing Process

The illustration in Figure 3 provides a detailed overview of the processing steps and highlights the key technical control points involved in the production of milk coffee. Manufacturing an aseptic milk coffee drink involves: [59] (i) coffee extraction, (ii) pH adjustment of the coffee extract, (iii) blending the coffee extract with sugar and milk, (iv) adding ingredients to milk coffee (emulsifiers, fragrance, etc.), (v) sterilizing with Ultra-high temperature instantaneous sterilization (UHT), (vi) aseptic filling after cooling.

### 4.2. Flavor-Preserving and Functional Processing Strategies

Consumers expect coffee drinks to possess both nutritional and functional qualities, without compromising their sensory experience. Therefore, it is imperative to advance milk coffee processing technology to ensure flavor consistency and lasting functionality. Cold-brew extraction has also been explored as a complementary low-temperature pathway within milk–coffee processing, with reports of smoother mouthfeel and higher retention of key volatiles and phenolics compared with hot extraction, providing a useful model for flavor and antioxidant stabilization in dairy matrices [10,13,60].

#### 4.2.1. Technology for Retaining the Flavor of Milk Coffee

BAS technology is used to improve the quality of milk coffee. Ikeda et al. [59] proposed a processing method for ready-to-drink milk coffee known as the BAS process. Following sterilization of milk and coffee, the milk and coffee extracts (Coffee extract 20.2%, Concentrated skim milk 13.8%, Water 57.7%) are mixed at 10 °C or lower, which effectively retains the original flavor of the milk coffee versus conventional blending-before-sterilization/UHT with pre-mix pH adjustment, BAS preserves retronasal coffee odorants and attenuates heat-driven flavor drift: in RTD milk coffee the total charm-value reached ~83% with BAS (no pH adjustment) versus ~56% under UHT/pH-adjusted blending [59], consistent with reports that higher thermal loads promote non-volatile staling markers in RTD coffee and heat-derived shifts in milk/coffee volatile fingerprints [61,62]. Operationally, UHT remains simpler and robust, whereas BAS places greater hygienic demands at the post-sterilization mixing step. Milk coffee processing also employs microencapsulation-based flavor recombination. Microencapsulation offers several advantages, including enhanced chemical and storage stability of flavorings (core materials), slower release to minimize gastrointestinal discomfort, improved flavor formulation, and stability in the presence of oxygen [63]. Kim et al. [64] studied microencapsulated flavored milk coffee. Caramel flavor compounds with antioxidant effects were introduced into the intestine and slowly released, whereas the coffee flavor was unaffected by oxygen.

Advances in flavor technology in milk processing technology allow consumers to conveniently enjoy classic milk coffee with a flavor comparable to that of freshly made coffee. At the same time, flavor technology can be used to create a variety of customizable flavors from classic to exotic to meet the tastes of different consumers.

#### 4.2.2. Technologies for Maintaining the Nutrition and Safety of Milk Coffee

Non-thermal technology is an emerging technology in the food industry [65]. It comprises PEF, high-voltage processing, ultrasound, pulsed light, and CP technologies [66]. Non-thermal technologies often preserve functional components such as proteins, minerals, and vitamins in food better than traditional thermal processing, while maintaining higher antioxidant quality [67]. And compared with UHT it achieves equivalent microbial safety at a much lower thermal load with smaller losses of key aroma volatiles and vitamins; however, because CP generates reactive species, strict control of working gas, discharge power, electrode geometry and exposure time is required to avoid off-flavors and packaging incompatibility [68]. The explicit application of non-thermal technology in milk coffee processing is mainly reflected in HPH. HPH acts on the milk-coffee system by applying high pressure, causing the rupture of fat globules and the dispersion of protein aggregates, which promotes the formation of a finer and more stable particle distribution in the system. Meanwhile, it affects the interaction between bioactive components and the matrix [37]. Ultrasonic technology employs high-frequency sound waves to disrupt the cell structure of microorganisms in semi-solid foods and liquid foods such as milk [65]. PEF can mitigate the detrimental effects of traditional thermal processing without affecting the nutritional, functional, and sensory properties of milk and dairy products [69]. Relative to UHT it typically yields smaller droplet size, a lower creaming index, a more negative zeta potential and higher phenolic retention during in vitro digestion, while practical constraints include higher specific energy use, greater equipment wear and the risk of pressure-induced protein unfolding outside optimal windows. Recent molecular-level studies combining molecular dynamics and spectroscopy have clarified how chlorogenic acids and related polyphenols associate with caseins, whey proteins, and lipid interfaces during processing and digestion, helping explain observed changes in polyphenol stability and bioaccessibility and guiding parameter selection for HPH, ultrasound, and PEF treatments [27,28,70].

Adding an appropriate amount of milk can enhance the safety of coffee. Acrylamide is a toxic compound that develops during the roasting process of coffee beans [71]. The acrylamide content in coffee decreases during storage by reacting with proteins and/or other substances to form acrylamide adducts. Yoshioka et al. [72] combined isotope labeling techniques with high-resolution mass spectrometry to detect acrylamide in canned milk coffee. Over a 4-month storage period, four adducts (3-hydroxy pyridine-acrylamide, pyridine-acrylamide adduct, lysine-acrylic acid, and cysteine-SO2-acrylic acid) decreased the total amount of acrylamide in canned milk coffee by 75.3%. This demonstrates that endogenous proteins contribute to acrylamide reduction in coffee beverages. Therefore, adding milk (about 10%) to coffee drinks can help improve the safety of milk-based coffee drinks.

Across processes summarized in Appendix A, we compare industrial readiness, throughput/cost–energy considerations, and regulatory/validation requirements for UHT, BAS, HPH, PEF, cold plasma, and power ultrasound to link technology choice with practical deployment. Briefly, UHT remains the high-TRL baseline with clear global frameworks but the highest thermal load; BAS preserves aroma via aseptic post-mixing yet requires stricter hygienic design and media-fill validation; HPH is broadly line-ready and cost-efficient for droplet refinement and stability; PEF shows growing beverage-scale adoption with competitive energy at target lethality but needs product-specific process validation; cold plasma is pilot-to-early industrial, achieving near-ambient inactivation while demanding tight control of discharge parameters and packaging compatibility; and power ultrasound provides dispersion/microstructure benefits with economics governed by power density and residence-time design [73,74,75,76,77,78]. Combining non-thermal technology with research on adding milk to reduce acrylamide content in coffee is expected to allow milk coffee to maintain its nutritional and sensory properties for a long time, reduce the use of preservatives and additives, and improve food safety. These technologies will meet consumers’ demand for green and healthy products, providing a more nutritious and safer consumption experience.

### 4.3. Natural Emulsifiers for the Texture and Stability of Milk Coffee

Natural and healthy emulsifiers and stabilizers, such as casein and whey proteins, are surface-active molecules that serve as protein-based emulsifiers and stabilizers. Milk protein is adsorbed at the oil-water interface, forming a protective layer around the oil droplets [78]. The mixed protein system, consisting of both milk and plant proteins, can serve as a novel food emulsifier. It exhibits physical stabilizing properties via adsorption and synergistic effects as well as the ability to disrupt protein aggregates [79]. For example, the electrostatic interaction between whey protein and soy protein forms a relatively stable interfacial layer, which can effectively protect the stability of the emulsion [80]. Sundaram et al. [81] stabilized instant coffee by incorporating bacterial cellulose and microcrystalline cellulose, which serve as a mechanical barrier to prevent fat-water agglomeration. The interaction between proteins and polysaccharides, as highlighted by Ribeiro et al. [82], improves emulsifying activity and creates a stable emulsion through covalent bonding. Based on the established understanding that emulsion stability is governed by interfacial layer formation and droplet interaction regulation [83], along with the specific adsorption behaviors of dairy and plant proteins at oil-water interfaces [79] and the enhanced stability achieved through protein-polysaccharide complexes [84,85], we have integrated these fundamental mechanisms into the unified schematic representation presented in Figure 5.

The innovative methods mentioned above present a perspective in food science: the use of bio-based polysaccharides, proteins, as emulsifiers to create stable food emulsions, which may lead to the development of more natural and sustainable food products with extended shelf life.

Therefore, research focusing on the nutritional and flavor aspects of milk coffee could lead to innovations and breakthroughs in areas such as microencapsulation, non-thermal processing, use of health food additives, replacement of vegetable fats with milk-based alternatives, and development of micromolecular water systems (low hertz water). Low hertz water describes water that has undergone a treatment method aimed at modifying its molecular structure or energy state through the application of low-frequency electromagnetic fields or vibrations. The term “hertz” (Hz) refers to cycles per second and is used to measure frequency. In this context, “low hertz” could imply that the treatment involves frequencies at the lower end of the electromagnetic spectrum, possibly in the range of kilohertz (kHz) or even lower. In addition, a critical review of the various processing techniques could identify potential research gaps and opportunities for innovation. Research into environmentally friendly, recyclable or biodegradable packaging materials is essential to reduce the environmental impact of milk coffee products.

### 4.4. Emerging Directions in Plant-Based Milk Coffee Systems

Product Features. Many dairy and non-dairy companies actively diversify their product portfolios using plant-based dairy alternatives in response to the global expansion of plant-based milk coffee [86]. Plant-based milk (soy, almond, oat, and rice) has a lower environmental impact than dairy products, which are more environmentally friendly due to efficient resource conversion and use of agricultural byproducts [87,88]. Furthermore, plant-based milks offer a variety of health benefits, such as providing essential nutrients including vitamins, minerals, healthy fats, and fiber while preventing lactose intolerance and milk allergies [89]. The global market for plant-based products is growing rapidly, and plant-based milks based on seeds such as grains, legumes, nuts, vegetables, and flax show immense potential in the coffee beverage industry.

Chung et al. [86] prepared 12 coffee samples using plant-based milk (oat, soy, almond, and coconut) and milk. Most consumers (53.8%) prefer milk coffee, while the rest (46.2%) were categorized as plant-based milk-coffee lovers. The key reasons for enjoying plant-based milk coffee samples include their smooth, milky, and thick qualities; however, sensory characteristics of plant-based milks typically include beany, earthy, bitter, spicy, astringent, sandy, and thin tastes. Developing milk-based coffee recipes or processing techniques with a milky, thick, and smooth flavor will help increase public awareness of plant-based milk.

Plant-based milk coffee products, including those based on oat, soy, almond, or pea, are gaining attention due to environmental and health considerations. However, they frequently suffer from phase instability and flavor imbalance when mixed with acidic coffee. Plant-based milk coffee undergoes phase separation. Soy milk in coffee often curdles due to factors like acidity, temperature, concentration, and mixing order. This incompatibility largely arises from electrostatic aggregation and polyphenol–protein complexation. Enzymatic modification and lactic fermentation can effectively alleviate these issues by increasing protein solubility, modulating surface charge, and generating smaller peptides or exopolysaccharides that improve phase stability. These treatments also enhance flavor retention by reducing excessive binding with phenolics and preserving volatile compounds during heating or storage [79,85,90]. Brown et al. [91] found this phase separation reversible, as cooling or adjusting soy milk concentration can restore a uniform mixture.

## 5. Future Perspectives

Milk coffee is an important component of coffee production. When combined with the results of the data survey on “coffee products,” four main trends were proposed for the development of innovative coffee products (Figure 6): (i) Health and Nutritional Enhancement: Milk coffee, added with yak butter, ghee, cheese, and other dairy products, is believed to offer nutritional value in providing energy as well as proteins, vitamins, minerals and other substances. Incorporating functional ingredients such as prebiotics, probiotics, and omega-3 fatty acids, chlorogenic acid can enhance the health profile of recombined milk coffee, making it a more nutritionally dense beverage. (ii) Eco-friendly plant-based milk coffee (oat milk coffee, or almond milk coffee). (iii) Sensory Experience (medicinal food milk coffee, rose milk coffee, wild mushroom milk coffee, hot area characteristic fruit and syrup, wine, tea, eggs, light milk, ice cream, cheese, yogurt). (iv) Youth-targeted Milk Coffee Options (mint factor cold milk coffee, dessert cream coffee appearance, milk coffee gel candy, sodium alginate embedded coffee burst beads, the integration of different countries/regions of the culture of milk coffee), used with yogurt, milk tea, ice cream, and bread and other different application scenarios. To advance these trends from concept to practice, priority research needs are multi-omics mapping of protein–polyphenol–lipid assemblies across roast level, brew type and milk matrix with links to sensory and stability [28,92]; thermodynamic and spectroscopic quantification of binding energetics and structural change under process-relevant pH, ionic strength and fat levels [28,93]; molecular-dynamics–guided models that locate binding sites and predict interface remodeling by temperature, shear and homogenization [94,95]; standardized INFOGEST-based in vitro–in vivo correlation to reconcile bioaccessibility versus bioavailability [96,97]; and scale-up enablers—process-analytical technologies, digital twins and comparative LCA/TEA—that benchmark BAS, HPH, PEF and CP against UHT [98,99]. Additionally, future work should validate these mechanisms in milk–coffee matrices matched to real formulations and processing histories by combining in vivo and ex vivo transport/kinetic approaches (e.g., epithelial transport models and plasma metabolite profiling) with multi-omics integration (metabolomics, proteomics, lipidomics), thereby establishing quantitative in vitro–in vivo correlations and mechanistic attribution [95].

In 2022, China’s coffee planting area was 86,300 hectares, with an output of 115,100 tons, ranking 14th globally, and the coffee consumption was 288,000 tons, ranking 7th in the world. Yunnan coffee accounts for over 98% of the country’s planting area, production, and agricultural output value, making Yunnan the largest coffee province in China [100]. Consequently, there is considerable importance in promoting the utilization of Yunnan coffee in the food industry. Specifically, it is recommended to (i) Yunnan coffee varieties Catimor and Burbon for milk coffee, and (ii) explore the functional components of coffee by-products for the creation of milk snack products. The intensive processing of coffee produces several by-products such as coffee husk pectin, coffee silver husk, and coffee grounds. Coffee processing by-products include caffeine, chlorogenic acid, alkaloids, coffee pectin, synaptic, and ferulic acids and other active molecular resources [18,101]. These by-products can be used to produce instant milk coffee powder rich in chlorogenic acid and caffeine. Moreover, innovative products such as milk-based snacks and milk-based coffee energy bars can be developed.

## 6. Conclusions

This article explores milk coffee, which has seen substantial market expansion, propelled by robust consumption and research efforts primarily in the United States and China. The sensory, nutritional, and functional attributes of milk coffee are intricately linked to the interactions among milk proteins, fats, and coffee polyphenols, which collectively shape its flavor profile, stability, and health-promoting qualities.

Research shows that protein–polyphenol complexes enhance antioxidant activity and stability, and their interactions are predominantly non-covalent (hydrophobic and hydrogen-bonding, with van der Waals contributions), exemplified by chlorogenic acid binding to α-lactalbumin/α-casein and to β-lactoglobulin/β-casein. These interactions can partially unfold proteins, lower surface hydrophobicity, and improve β-lactoglobulin solubility, antioxidant capacity, and thermal stability. The addition of milk to coffee mitigates astringency and modulates flavor release, with low-fat milk (0.5%) often better at preserving aroma; visual appearance can also subtly shift perceived sweetness. Polyphenols provide anti-inflammatory and antioxidant benefits, while milk addition differentially affects their bioaccessibility; decaffeinated coffee with milk may retain more antioxidants.

In the realm of innovative processing, technologies like microencapsulation, selected non-thermal sterilization, and natural emulsifiers offer promising avenues to enhance flavor, nutritional value, and shelf life. Additionally, plant-based milk alternatives, such as oat milk and soy milk, are gaining traction among consumers, albeit with challenges related to texture and stability.

Looking ahead, research should prioritize predictive, transferable rules that link processing parameters to interfacial architecture and product performance. Future research should quantify how specific routes tune protein–polyphenol–lipid assemblies, droplet structure, and interfacial rheology; derive process–structure–function models connecting binding modes to aroma release, astringency, physical stability, and phenolic bioaccessibility; standardize analytical protocols and reference matrices for cross-study comparability; optimize protein–fat–polyphenol ratios and fat-phase design under realistic manufacturing constraints; enhance plant-based compatibility via enzymatic or fermentation pretreatments and tailored emulsifier systems; and validate health relevance with standardized bioaccessibility and in vivo endpoints to develop new milk coffee products with excellent sensory experience, high nutritional functionality, and environmental sustainability. These efforts will promote the development of the industry in a more scientific and efficient direction. Together, these integrated priorities strengthen the bridge from mechanism to manufacturing (including the deployment of large-scale blending-after-sterilization systems) and enable credible, reproducible industrial adoption. This review underscores the existing gaps in milk coffee research, particularly in mechanistic clarity, method standardization, and translational validation, and advocates for leveraging advancements in processing technology and product innovation to cater to the ever-evolving demands of consumers and factory production.

## Figures and Tables

**Figure 1 foods-14-04043-f001:**
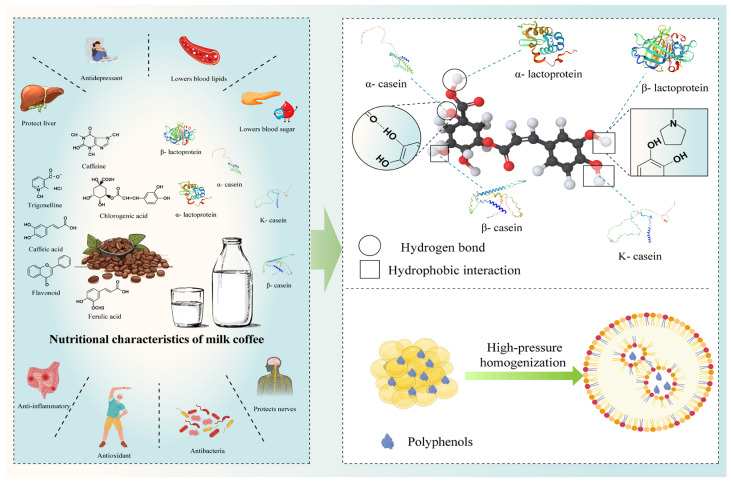
Nutritional and functional properties of milk and coffee and the interaction of the main proteins and fats in milk with coffee polyphenols.

**Figure 2 foods-14-04043-f002:**
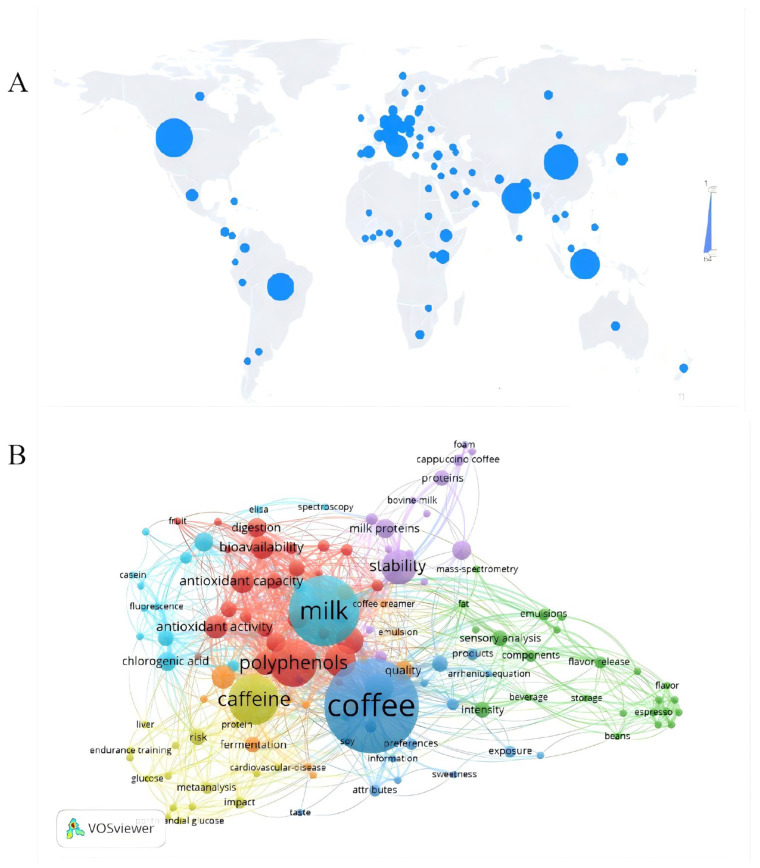
Recent trends in research on milk coffee. (**A**) Comparative analysis of articles published on milk coffee in different countries. Web of Science search for coffee products. (**B**) An overlay visualization of titles and abstracts of 324 selected references from 2014–2024.

**Figure 3 foods-14-04043-f003:**
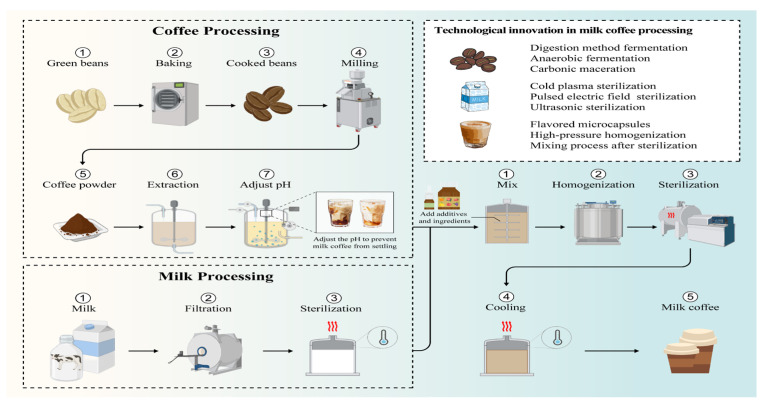
Milk coffee processing key control points and innovative processing methods.

**Figure 4 foods-14-04043-f004:**
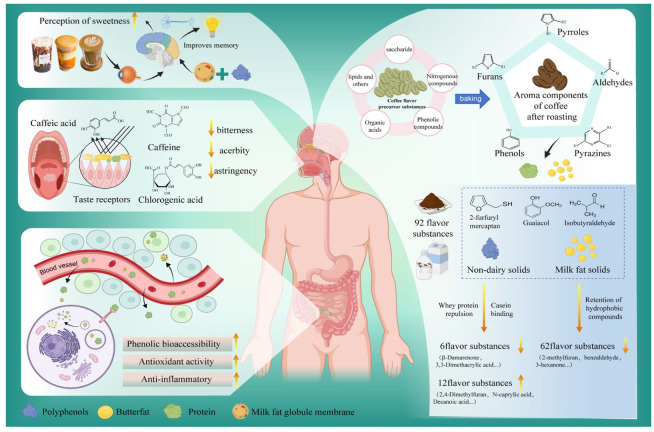
The influence of adding milk to coffee on human senses and its protective effect on coffee polyphenols in the gastrointestinal tract.

**Figure 5 foods-14-04043-f005:**
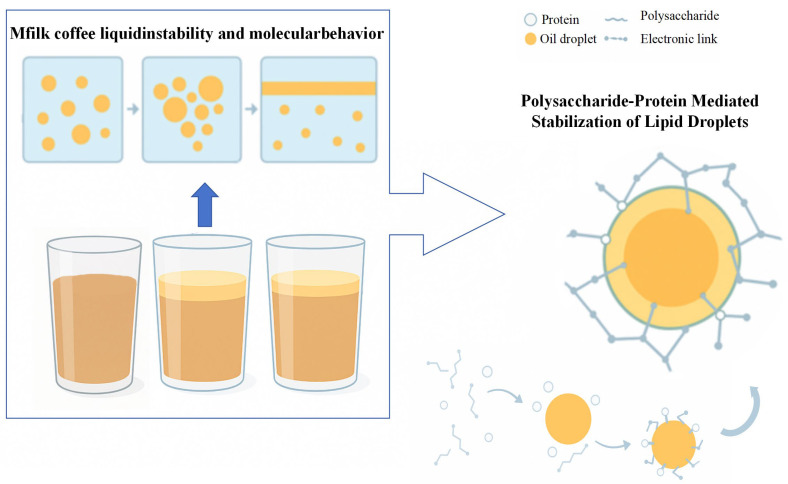
Schematic illustration of protein and polysaccharide stabilization mechanisms in milk coffee emulsions.

**Figure 6 foods-14-04043-f006:**
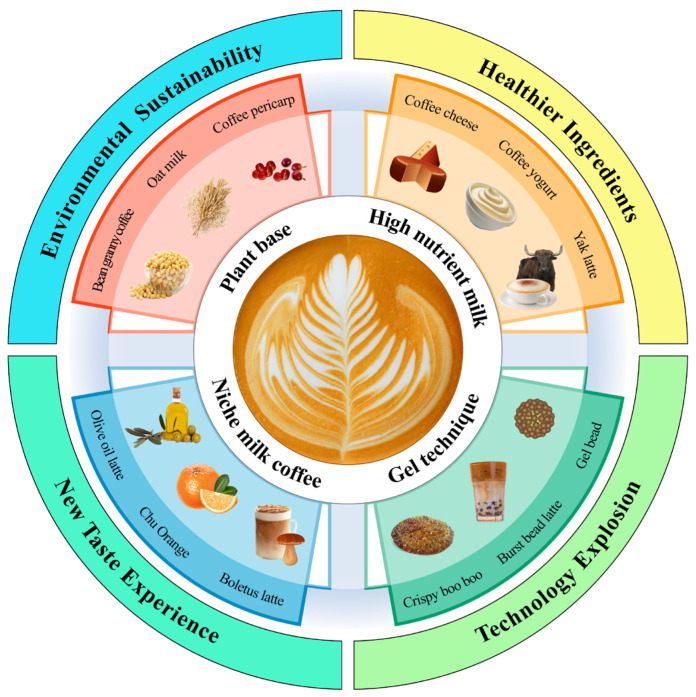
Milk coffee innovative products.

## Data Availability

No new data were created or analyzed in this study. Date sharing is not applicable.

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
