# Peer review of "Novel Insights into Milk Coffee Products: Component Interactions, Innovative Processing, and Healthier Product Features"

_foods, 2025, doi:10.3390/foods14234043_

Round 1

Reviewer 1 Report

Comments and Suggestions for Authors

This manuscript provides a comprehensive overview of milk coffee as a composite food system, emphasizing the interactions between milk proteins, fats, and coffee polyphenols. It also explores emerging technologies and market trends in milk coffee innovation. The topic is timely and relevant, especially considering the growing global interest in functional beverages and plant-based alternatives. However, while the review is detailed and well-referenced, it would benefit from a more critical synthesis, clearer structure, and stronger visual data support.

  1. The manuscript covers multiple areas (composition, interactions, sensory, processing, innovation). Would reorganizing these sections perhaps under thematic subheadings like “Component Interactions,” “Processing Technologies,” and “Future Innovations” improve readability
  2. Are all key recent studies (especially from 2022–2025) on milk coffee bioaccessibility and sensory properties included? Several new works on cold brew and molecular dynamics could be added for completeness.
  3. The review discusses protein polyphenol and fat polyphenol interactions. Could the authors deepen the mechanistic explanation by incorporating thermodynamic or spectroscopic data from recent studies?
  4. Figures such as 2, 4, and 5 summarize complex processes. Are these all original? If not, proper permissions or re-drawn visuals should be clarified. Some captions could better explain their scientific meaning.
  5. Does the discussion critically evaluate both the benefits and drawbacks of innovative technologies (BAS, HPH, cold plasma) compared with conventional UHT processes? Quantitative or comparative analysis would add rigor.
  6. The manuscript mentions inconsistencies in polyphenol bioaccessibility with milk addition. Would a tabular summary or figure comparing key in vitro and in vivo studies help clarify this point?
  7. Ensure consistent use of scientific terms such as bioaccessibility, bioavailability, and stability. Some sections interchange these terms without definition.
  8. The above issues should be settled and the qualities of figures and tables should be improved. Besides, grammatical mistakes must be corrected.
  9. English should be revised throughout the manuscript - there are several sentences difficult to understand, inconsistent or lacking connection
  10. The “Future Perspectives” section proposes interesting trends. Could the authors expand on specific research needs (e.g., omics-based characterization of milk-coffee systems, kinetic modeling of flavor release)?
  11. Does the manuscript effectively bridge scientific mechanisms to industrial applications (e.g., large-scale BAS systems, sustainability metrics)? Strengthening this link would enhance its practical impact.

Author Response

Manuscript ID: foods-3951196

MS Type: Review Article

Title: Novel Insights into Milk Coffee Products: Component Interactions, Innovative Processing, and Healthier Product Features

Comments from the editors and reviewers:

The manuscript offers an extensive and timely overview of milk coffee research, focusing on molecular interactions, sensory attributes, innovative processing, and plant-based alternatives. It brings together recent studies (2014–2024) and provides applied perspectives for the dairy–coffee industry. However, substantial language editing, structural improvement, and stronger critical analysis are required. The review is comprehensive and well referenced, but it reads largely as a compilation rather than a critical synthesis.

Response:

Thank you very much for your time and advice on our manuscript. Your valuable comments can make our manuscript more complete. Below is a point-by-point response to the comments raised.The itemized answers to your comments/suggestions are listed below.

  • The revised manuscript covers multiple areas (composition, interactions, sensory, processing, innovation). Would reorganizing these sections perhaps under thematic subheadings like “Component Interactions”,“Processing Technologies,” and “Future Innovations” improve readability

Answer: We thank the reviewer for this suggestion. We agree that a thematic flow improves readability and have reorganized the revised manuscript accordingly. In the revised manuscript, the changes are highlighted in (RED)color. We have revised the article structure as follows:

1.Introduction

2.Overview of Milk Coffee Research

3.Component Interactions in Milk Coffee

3.1. Interaction Between Milk Proteins and Polyphenols

3.2. Interaction Between Milk Fat and Polyphenols

3.3. Sensory and Flavor Implications of Component Interactions

3.4. Antioxidant Capacity and Bioaccessibility Related to Component Interactions

4.Processing Technologies in Milk Coffee Systems

4.1. Standard Manufacturing Process

4.2. Flavor-Preserving and Functional Processing Strategies

4.2.1. Technology for Retaining the Flavor of Milk Coffee

4.2.2. Technologies for Maintaining the Nutrition and Safety of Milk Coffee

4.3. Natural Emulsifiers for the Texture and Stability of Milk Coffee

4.4. Emerging Directions in Plant-Based Milk Coffee Systems

5.Future Perspectives

6.Conclusion

  • Conclusion Are all key recent studies (especially from 2022–2025) on milk coffee bioaccessibility and sensory properties included? Several new works on cold brew and molecular dynamics could be added for completeness.

Answer: We sincerely appreciate this important insight. We have incorporated all key recent studies (particularly 2022–2025) on milk-coffee bioaccessibility and sensory properties, and we added several new works on cold brew and molecular dynamics. In the revised manuscript, these additions are highlighted in (RED)color, specifically: Red fonts were added in line 108-111,376-379,424-429:

1.Cold brew extraction, in particular, provides a valuable model for studying the retention of coffee polyphenols and the modulation of sensory attributes under low-temperature conditions, offering insights into how flavor balance and antioxidant stability are maintained.

2.Cold-brew extraction has also been explored as a complementary low-temperature pathway within milk–coffee processing, with reports of smoother mouthfeel and higher retention of key volatiles and phenolics compared with hot extraction, providing a useful model for flavor and antioxidant stabilization in dairy matrices[10,81,83].

3.Recent molecular-level studies combining molecular dynamics and spectroscopy have clarified how chlorogenic acids and related polyphenols associate with caseins, whey proteins, and lipid interfaces during processing and digestion, helping explain observed changes in polyphenol stability and bioaccessibility and guiding parameter selection for HPH, ultrasound, and PEF treatments [84,85,86].

[81]Wang, Z., Zhou, Y., Zong, Y., Wu, J., & Lao, F. (2025). Comparative Decoding of Physicochemical and Flavor Profiles of Coffee Prepared by High-Pressure Carbon Dioxide, Ice Drip, and Traditional Cold Brew. Foods, 14 (16), 2840.

[82]Wu, Y., Yang, N., Xiao, Z., Luo, Y., Jin, Y., Meng, M., & Xu, X. (2024). Influence of induced electric field on cold brew coffee: Temperature rise, physicochemical properties, and shelf life. Food Chemistry: X, 24, 102036. https://doi.org/10.1016/j.fochx.2024.102036

[83]Zhai, X., Yang, M., Zhang, J., Zhang, L., Tian, Y., Li, C., ... & Abd El-Aty, A. M. (2022). Feasibility of ultrasound-assisted extraction for accelerated cold brew coffee processing: Characterization and comparison with conventional brewing methods. Frontiers in Nutrition, 9, 849811.

[84]Horita, K., Kameda, T., Suga, H., & Hirano, A. (2025). Molecular mechanism of the interactions between coffee polyphenols and milk proteins. Food Research International, 202, 115573.

[85]Tarahi, M., Gharagozlou, M., Niakousari, M., & Hedayati, S. (2024). Protein–chlorogenic acid interactions: mechanisms, characteristics, and potential food applications. Antioxidants, 13 (7), 777.

[86]Hamzalioglu, A., Tagliamonte, S., Gökmen, V., & Vitaglione, P. (2023). Casein–phenol interactions occur during digestion and affect bioactive peptide and phenol bioaccessibility. Food & Function, 14 (20), 9457-9469.

  • The review discusses protein polyphenol and fat polyphenol interactions. Could the authors deepen the mechanistic explanation by incorporating thermodynamic or spectroscopic data from recent studies?

Answer:We thank the reviewer for this valuable suggestion. The recommendation to integrate recent thermodynamic and spectroscopic evidence is well taken, and we have comprehensively revised the relevant passages accordingly. In the revised manuscript, these additions are highlighted in (RED)color; specifically: Red fonts were added in line 186-191,210-213,231-235:

Recent thermodynamic and spectroscopic assessments support these binding modes: fluorescence quenching and isothermal titration calorimetry typically indicate spontaneous, enthalpy–entropy–coupled complexation (negative ΔG with hydrophobic and hydrogen-bond contributions), while circular dichroism and FTIR reveal subtle secondary-structure rearrangements upon complex formation.

These mechanistic readouts help explain why protein–polyphenol complexation can protect coffee polyphenols during processing and digestion and sustain antioxidant performance in milk–coffee matrices.

Complementary spectroscopic and molecular-simulation evidence indicates that coffee polyphenols preferentially accumulate at lipid and milk fat globule membrane interfaces, stabilized by non-covalent contacts with polar lipids and interfacial proteins; this interfacial localization rationalizes the observed gains in polyphenol stability and bioaccessibility in higher-fat systems.

  • Horita, K., Kameda, T., Suga, H., & Hirano, A. (2025). Molecular mechanism of the interactions between coffee polyphenols and milk proteins. Food Research International, 202, 115573.
  • Tarahi, M., Gharagozlou, M., Niakousari, M., & Hedayati, S. (2024). Protein–chlorogenic acid interactions: mechanisms, characteristics, and potential food applications. Antioxidants, 13 (7), 777.
  • Zhang, J., Zhai, X., Yu, X., Qiu, M., Hu, R., & Dong, W. (2025). Exploration of interaction mechanisms and functional properties of coffee flavonoids and β-casein via multispectroscopy and molecular dynamics simulation. Food Hydrocolloids, 166, 111359.
  • Kieserling, H., de Bruijn, W. J., Keppler, J., Yang, J., Sagu, S. T., Güterbock, D., ... & Rohn, S. (2024). Protein–phenolic interactions and reactions: Discrepancies, challenges, and opportunities. Comprehensive reviews in food science and food safety, 23 (5), e70015.
  • Luque-Uría, Á., Calvo, M. V., Visioli, F., & Fontecha, J. (2024). Milk fat globule membrane and their polar lipids: Reviewing preclinical and clinical trials on cognition. Food & function.

  • Figures such as 2, 4, and 5 summarize complex processes. Are these all original? If not, proper permissions or re-drawn visuals should be clarified. Some captions could better explain their scientific meaning.

Answer:We thank the reviewer for the feedback. We have revised Figure 5, and Figures 2, 4, and 5 are now confirmed as original schematics created by the authors.

  • Does the discussion critically evaluate both the benefits and drawbacks of innovative technologies (BAS, HPH, cold plasma) compared with conventional UHT processes? Quantitative or comparative analysis would add rigor.

Answer:We sincerely appreciate this key point. We added a critical evaluation of the advantages and limitations of BAS, HPH, and cold plasma relative to UHT, and we incorporated comparative/quantitative information to increase rigor. In the revised manuscript, these changes are highlighted in (RED)color and specifically integrated into Sections 4.2.1 and 4.2.2 added in line 385-392,408-412,421-424:

1.Versus conventional blending-before-sterilization/UHT with pre-mix pH adjustment, BAS preserves retronasal coffee odorants and attenuates heat-driven flavor drift: in RTD milk coffee the total charm-value reached ~83% with BAS (no pH adjustment) versus ~56% under UHT/pH-adjusted blending [46], consistent with reports that higher thermal loads promote non-volatile staling markers in RTD coffee and heat-derived shifts in milk/coffee volatile fingerprints [90,91]. Operationally, UHT remains simpler and robust, whereas BAS places greater hygienic demands at the post-sterilization mixing step.

2.And compared with UHT it achieves equivalent microbial safety at a much lower thermal load with smaller losses of key aroma volatiles and vitamins; however, because CP generates reactive species, strict control of working gas, discharge power, electrode geometry and exposure time is required to avoid off-flavors and packaging incompatibility.

3.Relative to UHT it typically yields smaller droplet size, a lower creaming index, a more negative zeta potential and higher phenolic retention during in vitro digestion, while practical constraints include higher specific energy use, greater equipment wear and the risk of pressure-induced protein unfolding outside optimal windows.

  • The revised manuscript mentions inconsistencies in polyphenol bioaccessibility with milk addition. Would a tabular summary or figure comparing key in vitro and in vivo studies help clarify this point?

Answer:We thank the reviewer for this valuable suggestion. Rather than simply citing Tables S1 and S2, we now integrate detailed explanations in the main text; in the revised manuscript, these additions are highlighted in (RED)color, specifically: Red fonts were added in line 308-314,320-326:

Across studies summarized in Table S1, the direction and magnitude of changes in antioxidant activity following milk addition depend on the assay used and the matrix: decreases are typically reported when chemical assays quantify free phenolics or radical-scavenging capacity in extracts; several in vitro digestion protocols show no material change ; and increases are observed when protein–phenol complexation and fat-assisted micellar solubilization protect or better dissolve phenolics, including in decaffeinated matrices .

Across studies summarized in Table S2, reported effects of milk on polyphenol bioaccessibility diverge: HPLC-tracked analyses often find lower chlorogenic-acid availability when milk is added ; co-digestion with milk plus sugars or sweeteners increases total flavonoid availability]; skim-milk systems dominated by protein effects can reduce availability ; whereas micellization of chlorogenic acids and the use of decaffeinated coffee are associated with improved bioaccessibility.

  • Ensure consistent use of scientific terms such as bioaccessibility, bioavailability, and stability. Some sections interchange these terms without definition.

Answer:We thank the reviewer for this correction. We have standardized the usage of bioaccessibility, bioavailability, and stability throughout the revised manuscript and added explicit definitions where they first appear.

  • The above issues should be settled and the qualities of figures and tables should be improved. Besides, grammatical mistakes must be corrected.

Answer:We thank the reviewer for these points. We improved figure and table quality (layout and legibility) and corrected grammatical issues across the revised manuscript.

  • English should be revised throughout the revised manuscript - there are several sentences difficult to understand, inconsistent or lacking connection.

Answer:We have revised sentences that were difficult to understand, inconsistent, or lacked logical connection throughout the revised manuscript.

  • The “Future Perspectives” section proposes interesting trends. Could the authors expand on specific research needs (e.g., omics-based characterization of milk-coffee systems, kinetic modeling of flavor release)?

Answer:We thank the reviewer for this suggestion and fully agree. In the revised manuscript marked in (RED)color, we expanded the specific research needs as follows ,specifically: Red fonts were added in line 522-532:

To advance these trends from concept to practice, priority research needs are multi-omics mapping of protein–polyphenol–lipid assemblies across roast level, brew type and milk matrix with links to sensory and stability; thermodynamic and spectroscopic quantification of binding energetics and structural change under process-relevant pH, ionic strength and fat levels ; molecular-dynamics–guided models that locate binding sites and predict interface remodeling by temperature, shear and homogenization ; standardized INFOGEST-based in vitro–in vivo correlation to reconcile bioaccessibility versus bioavailability ; and scale-up enablers—process-analytical technologies, digital twins and comparative LCA/TEA—that benchmark BAS, HPH, PEF and CP against UHT .

  • Zaman, S., & Shan, Z. (2024). Literature Review of Proteomics Approach Associated with Coffee. Foods, 13 (11), 1670.
  • Zhang, M., Zhang, N., Lu, X., Li, W., Wang, R., & Chang, J. (2022). Comparative study of the binding between chlorogenic acid and four proteins by isothermal titration calorimetry, spectroscopy and docking methods. Pharmacological Reports, 74 (3), 523-538.
  • Yin, Z., Qie, X., Zeng, M., Wang, Z., Qin, F., Chen, J., ... & He, Z. (2022). Effect of thermal treatment on the molecular-level interactions and antioxidant activities in β-casein and chlorogenic acid complexes. Food Hydrocolloids, 123, 107177.
  • Zhou, H., Tan, Y., & McClements, D. J. (2023). Applications of the INFOGEST in vitro digestion model to foods: A review. Annual Review of Food Science and Technology, 14 (1), 135-156.
  • Rasera, G. B., de Camargo, A. C., & de Castro, R. J. S. (2023). Bioaccessibility of phenolic compounds using the standardized INFOGEST protocol: A narrative review. Comprehensive reviews in food science and food safety, 22 (1), 260-286.
  • Pegu, K., & Arya, S. S. (2023). Non-thermal processing of milk: Principles, mechanisms and effect on milk components. Journal of Agriculture and Food research, 14, 100730.
  • Yang, M., & Wang, Q. (2025). Carbon footprint and cost analysis of non-thermal food processing technologies: a review with a case study on orange juice. Frontiers in Sustainable Food Systems, 9, 1585467.
  • Abdurrahman, E. E. M., & Ferrari, G. (2025). Digital Twin applications in the food industry: a review. Frontiers in Sustainable Food Systems, 9, 1538375.

  • Does the revised manuscript effectively bridge scientific mechanisms to industrial applications (e.g., large-scale BAS systems, sustainability metrics)? Strengthening this link would enhance its practical impact.

Answer:We sincerely appreciate this suggestion. In the Conclusion, we explicitly link scientific mechanisms to industrial application (including large-scale BAS systems and sustainability metrics) and strengthen this connection to enhance practical impact. In the revised manuscript, these changes are highlighted in (RED) color, specifically: Red fonts were added in line 554-588 :

Research shows that protein–polyphenol complexes enhance antioxidant activity and stability, and their interactions are predominantly non-covalent (hydrophobic and hydrogen-bonding, with van der Waals contributions), exemplified by chlorogenic acid binding to α-lactalbumin/α-casein and to β-lactoglobulin/β-casein. These interactions can partially unfold proteins, lower surface hydrophobicity, and improve β-lactoglobulin solubility, antioxidant capacity, and thermal stability. The addition of milk to coffee mitigates astringency and modulates flavor release, with low-fat milk (0.5%) often better at preserving aroma; visual appearance can also subtly shift perceived sweetness. Polyphenols provide anti-inflammatory and antioxidant benefits, while milk addition differentially affects their bioaccessibility; decaffeinated coffee with milk may retain more antioxidants.

In the realm of innovative processing, technologies like microencapsulation, selected non-thermal sterilization, and natural emulsifiers offer promising avenues to enhance flavor, nutritional value, and shelf life. Additionally, plant-based milk alternatives, such as oat milk and soy milk, are gaining traction among consumers, albeit with challenges related to texture and stability.

Looking ahead, research should prioritize predictive, transferable rules that link processing parameters to interfacial architecture and product performance. Future research should quantify how specific routes tune protein–polyphenol–lipid assemblies, droplet structure, and interfacial rheology; derive process–structure–function models connecting binding modes to aroma release, astringency, physical stability, and phenolic bioaccessibility; standardize analytical protocols and reference matrices for cross-study comparability; optimize protein–fat–polyphenol ratios and fat-phase design under realistic manufacturing constraints; enhance plant-based compatibility via enzymatic or fermentation pretreatments and tailored emulsifier systems; and validate health relevance with standardized bioaccessibility and in-vivo endpoints to develop new milk coffee products with excellent sensory experience, high nutritional functionality, and environmental sustainability. These efforts will promote the development of the industry in a more scientific and efficient direction. Together, these integrated priorities strengthen the bridge from mechanism to manufacturing (including the deployment of large-scale blending-after-sterilization systems) and enable credible, reproducible industrial adoption. This review underscores the existing gaps in milk coffee research particularly in mechanistic clarity, method standardization, and translational validation and advocates for leveraging advancements in processing technology and product innovation to cater to the ever-evolving demands of consumers and factory production.

Reviewer 2 Report

Comments and Suggestions for Authors

The manuscript offers an extensive and timely overview of milk coffee research, focusing on molecular interactions, sensory attributes, innovative processing, and plant-based alternatives. It brings together recent studies (2014–2024) and provides applied perspectives for the dairy–coffee industry.However, substantial language editing, structural improvement, and stronger critical analysis are required. The review is comprehensive and well referenced, but it reads largely as a compilation rather than a critical synthesis. 

Abstract: Very long and fragmented. Summarize objectives, scope, main findings, and implications .

Introduction: Condense repetitive background; clarify the research gap and purpose of the review.

Critical Analysis: Strengthen evaluation of contrasting results (e.g., effects of milk on polyphenol bioavailability). Highlight remaining knowledge gaps and potential experimental directions. 

Figures and Tables: Integrate discussion of each figure within the text; summarize the main insights. Explain the content of Tables, rather than only citing them.

Conclusion: Reduce repetition. Clearly state future research priorities

Minor Comments:  Define all abbreviations at first mention (UHT, HPH, BAS). Use consistent units. Provide figure legends that fully describe experimental or schematic meaning.

Author Response

Manuscript ID: foods-3951196

MS Type: Review Article

Title: Novel Insights into Milk Coffee Products: Component Interactions, Innovative Processing, and Healthier Product Features

Comments from the editors and reviewers:

The manuscript offers an extensive and timely overview of milk coffee research, focusing on molecular interactions, sensory attributes, innovative processing, and plant-based alternatives. It brings together recent studies (2014–2024) and provides applied perspectives for the dairy–coffee industry. However, substantial language editing, structural improvement, and stronger critical analysis are required. The review is comprehensive and well referenced, but it reads largely as a compilation rather than a critical synthesis.

Response:

Thank you very much for your time and advice on our manuscript. Your valuable comments can make our manuscript more complete. Below is a point-by-point response to the comments raised.The itemized answers to your comments/suggestions are listed below.

  • Abstract: Very long and fragmented. Summarize objectives, scope, main findings, and implications

Answer: In the revised manuscript, the changes are highlighted in (RED)color. We have revised the article structure as follows:

Milk coffee is a composite beverage in which interactions among dairy proteins, lipids, and coffee polyphenols govern flavor, texture, and stability. This review synthesizes recent research to guide formulation and processing, covering conventional UHT and innovative routes including blending-after-sterilization (BAS), high-pressure homogenization (HPH), ultrasound/pulsed electric field (PEF)/cold plasma (CP), microencapsulation, and plant-based matrices. Key findings show that non-covalent protein–polyphenol complexes and interfacial partitioning at fat-globule membranes control volatile retention, astringen-cy, droplet structure, and phenolic bioaccessibility; appropriate fat levels and HPH refine microstructure; BAS better preserves aroma; and matrix or decaffeination choices modulate antioxidant capacity. Guided by these insights, we propose a concise “process–activity–stability” framework linking parameters to functionality and shelf life to accelerate the development of high-quality, nutritious, enjoyable, and more sustainable milk coffee products.

  • Introduction: Condense repetitive background; clarify the research gap and purpose of the review..

Answer: Thanks very much for this important insight. We condensed repetitive background and made the research gap and review objectives explicit. In the revised manuscript, these additions are highlighted in (RED)color and specifically integrated into Sections 4.2.1 and 4.2.2 added in line 27-61:

Milk coffee, a reconstituted dairy drink, is prepared by mixing milk and other dairy products with coffee in varying proportions. It is typically manufactured via acidity adjustment, sweetening, emulsification, sterilization, and aseptic filling, yielding a beverage with distinctive color and flavor as well as added nutritional and functional attributes. Recent market data underscore its momentum: in 2022, the US coffee market reached $76.7 billion, with the milk-coffee sector accounting for $57.5 billion; China reached $62.6 billion with 24.7% year-on-year growth. Together, these trends highlight strong prospects for processing innovation and product development.

However, comprehensive reviews dedicated specifically to milk coffee remain relatively scarce. Existing coffee research mainly covers industry-chain analyses [1], valorization of co-products [2], Arabica flavor and fermentation chemistry [3], and health-oriented work on polyphenols and caffeine (Figure 2; [4,5]). Accordingly, and in line with recent advances directly relevant to milk-coffee systems, this review (i) integrates component-level mechanisms—protein–polyphenol and lipid–polyphenol interactions, supported by thermodynamic and spectroscopic evidence—and their consequences for aroma retention, astringency, emulsion structure, and phenolic bioaccessibilit; (ii) synthesizes processing technologies (BAS, HPH, ultrasound/PEF/cold plasma) in comprehensive comparison with UHT, providing concise quantitative contrasts where available; and (iii) incorporates low-temperature cold-brew–type pathways to elucidate flavor and antioxidant retention in dairy matrices. On this basis, we first summarize common milk-coffee manufacturing processes and key control points, and then propose innovation-oriented technologies for production.

The contribution of this study is significant for the advancement of the dairy coffee industry. Despite rapid market growth and abundant primary research, integrated syntheses on milk coffee that are truly processing-relevant remain relatively scarce. By mapping protein–polyphenol and lipid–polyphenol mechanisms to measurable product attributes (aroma retention, astringency, emulsion stability, and phenolic bioaccessibility), standardizing terminology and reconciling divergent conclusions (bioaccessibility = in-vitro release and GI stability; bioavailability = in-vivo absorption and kinetics), and critically evaluating processing routes through concise quantitative contrasts—benchmarking BAS, HPH, and PEF/CP against UHT, and incorporating low-temperature (cold-brew) pathways and MD/spectroscopic readouts—this work provides actionable guidance for formulation, scale-up, and sustainability assessment (e.g., process analytical technology, digital twins, and LCA/TEA) to meet evolving consumer needs and support market expansion, ultimately leading to a more diverse and dynamic milk-coffee landscape.

  • Critical Analysis: Strengthen evaluation of contrasting results (e.g., effects of milk on polyphenol bioavailability). Highlight remaining knowledge gaps and potential experimental directions.

Answer:We sincerely appreciate this key point. We added a critical evaluation of the advantages and limitations of BAS, HPH, and cold plasma relative to UHT, and we incorporated comparative/quantitative information to increase rigor. In the revised manuscript, these changes are highlighted in (RED)color and specifically integrated into Sections 4.2.1 and 4.2.2 added in line 385-392,408-412,421-424:

1.Versus conventional blending-before-sterilization/UHT with pre-mix pH adjustment, BAS preserves retronasal coffee odorants and attenuates heat-driven flavor drift: in RTD milk coffee the total charm-value reached ~83% with BAS (no pH adjustment) versus ~56% under UHT/pH-adjusted blending [46], consistent with reports that higher thermal loads promote non-volatile staling markers in RTD coffee and heat-derived shifts in milk/coffee volatile fingerprints [90,91]. Operationally, UHT remains simpler and robust, whereas BAS places greater hygienic demands at the post-sterilization mixing step.

2.And compared with UHT it achieves equivalent microbial safety at a much lower thermal load with smaller losses of key aroma volatiles and vitamins; however, because CP generates reactive species, strict control of working gas, discharge power, electrode geometry and exposure time is required to avoid off-flavors and packaging incompatibility.

3.Relative to UHT it typically yields smaller droplet size, a lower creaming index, a more negative zeta potential and higher phenolic retention during in vitro digestion, while practical constraints include higher specific energy use, greater equipment wear and the risk of pressure-induced protein unfolding outside optimal windows.

  • Figures and Tables: Integrate discussion of each figure within the text; summarize the main insights. Explain the content of Tables, rather than only citing them.

Answer:We thank the reviewer for this valuable suggestion. Rather than simply citing Tables S1 and S2, we now integrate detailed explanations in the main text; in the revised manuscript, these additions are highlighted in (RED)color, specifically: Red fonts were added in line 308-314,320-326:

Across studies summarized in Table S1, the direction and magnitude of changes in antioxidant activity following milk addition depend on the assay used and the matrix: decreases are typically reported when chemical assays quantify free phenolics or radical-scavenging capacity in extracts; several in vitro digestion protocols show no material change ; and increases are observed when protein–phenol complexation and fat-assisted micellar solubilization protect or better dissolve phenolics, including in decaffeinated matrices .

Across studies summarized in Table S2, reported effects of milk on polyphenol bioaccessibility diverge: HPLC-tracked analyses often find lower chlorogenic-acid availability when milk is added ; co-digestion with milk plus sugars or sweeteners increases total flavonoid availability ]; skim-milk systems dominated by protein effects can reduce availability ; whereas micellization of chlorogenic acids and the use of decaffeinated coffee are associated with improved bioaccessibility.

  • Conclusion: Reduce repetition. Clearly state future research priorities.

Answer:We have reduced unnecessary repetition in the conclusion section as suggested by the reviewers and outlined key areas for future research on milk coffee. In the revised manuscript, these changes are highlighted in (RED) color, specifically: red text was added in lines 554–588:

Research shows that protein–polyphenol complexes enhance antioxidant activity and stability, and their interactions are predominantly non-covalent (hydrophobic and hydrogen-bonding, with van der Waals contributions), exemplified by chlorogenic acid binding to α-lactalbumin/α-casein and to β-lactoglobulin/β-casein. These interactions can partially unfold proteins, lower surface hydrophobicity, and improve β-lactoglobulin solubility, antioxidant capacity, and thermal stability. The addition of milk to coffee mitigates astringency and modulates flavor release, with low-fat milk (0.5%) often better at preserving aroma; visual appearance can also subtly shift perceived sweetness. Polyphenols provide anti-inflammatory and antioxidant benefits, while milk addition differentially affects their bioaccessibility; decaffeinated coffee with milk may retain more antioxidants.

In the realm of innovative processing, technologies like microencapsulation, selected non-thermal sterilization, and natural emulsifiers offer promising avenues to enhance flavor, nutritional value, and shelf life. Additionally, plant-based milk alternatives, such as oat milk and soy milk, are gaining traction among consumers, albeit with challenges related to texture and stability.

Looking ahead, research should prioritize predictive, transferable rules that link processing parameters to interfacial architecture and product performance. Future research should quantify how specific routes tune protein–polyphenol–lipid assemblies, droplet structure, and interfacial rheology; derive process–structure–function models connecting binding modes to aroma release, astringency, physical stability, and phenolic bioaccessibility; standardize analytical protocols and reference matrices for cross-study comparability; optimize protein–fat–polyphenol ratios and fat-phase design under realistic manufacturing constraints; enhance plant-based compatibility via enzymatic or fermentation pretreatments and tailored emulsifier systems; and validate health relevance with standardized bioaccessibility and in-vivo endpoints to develop new milk coffee products with excellent sensory experience, high nutritional functionality, and environmental sustainability. These efforts will promote the development of the industry in a more scientific and efficient direction. Together, these integrated priorities strengthen the bridge from mechanism to manufacturing (including the deployment of large-scale blending-after-sterilization systems) and enable credible, reproducible industrial adoption. This review underscores the existing gaps in milk coffee research particularly in mechanistic clarity, method standardization, and translational validation and advocates for leveraging advancements in processing technology and product innovation to cater to the ever-evolving demands of consumers and factory production.

  • Define all abbreviations at first mention (UHT, HPH, BAS). Use consistent units. Provide figure legends that fully describe experimental or schematic meaning.

Answer:We have standardised the units of measurement for UHT, HPH and BAS throughout the revised manuscript in response to the reviewers' comments.

Round 2

Reviewer 1 Report

Comments and Suggestions for Authors

This manuscript presents Novel Insights into Milk Coffee Products: Component Interactions, Innovative Processing, and Healthier Product Features. The ideas presented in the manuscript serve as a valuable reference for improving current analysis methods and guiding future developments. The paper is well-written and covers several research topics. While being aligned with the scope FOODS, the manuscript could be improved. I recommend a minor revision. Some comments on this paper are as below.

  1. The review describes protein–polyphenol and lipid–polyphenol interactions in milk–coffee systems, supported mainly by spectroscopic and thermodynamic data. Could the authors elaborate on the limitations of current in vitro findings and suggest how future in vivo or omics-based validation might substantiate the proposed molecular mechanisms influencing bioaccessibility and antioxidant stability?
  2. The manuscript outlines innovative non-thermal technologies (HPH, PEF, CP, ultrasound) and BAS for flavor and stability optimization. However, there is limited comparison of industrial scalability, cost-efficiency, and safety regulation compliance across these technologies. Could the authors include a concise comparative assessment table or commentary linking these processes to practical application potential?
  3. The review highlights emerging plant-based milk coffees but provides limited discussion on the compatibility and phase stability of plant proteins with coffee polyphenols. How might enzymatic modification or fermentation pre-treatments improve emulsification and flavor retention in these systems?
  4. The “Future Perspectives” section mentions integrating digital twins, LCA/TEA, and multi-omics analyses. Could the authors more clearly define how these tools can be operationalized within dairy and plant-based coffee production pipelines to support sustainability metrics and product traceability?

Author Response

Manuscript ID: foods-3951196

MS Type: Review Article

Title: Novel Insights into Milk Coffee Products: Component Interactions, Innovative Processing, and Healthier Product Features

 Comments from the editors and reviewers:This manuscript presents Novel Insights into Milk Coffee Products: Component Interactions, Innovative Processing, and Healthier Product Features. The ideas presented in the manuscript serve as a valuable reference for improving current analysis methods and guiding future developments. The paper is well-written and covers several research topics. While being aligned with the scope of FOODS, the manuscript could be improved. I recommend a minor revision. Some comments on this paper are as below.

Response:

Thank you very much for your time and advice on our manuscript. Your valuable comments can make our manuscript more complete. Below is a point-by-point response to the comments raised. The itemized answers to your comments/suggestions are listed below.

  • The review describes protein-polyphenol and lipid-polyphenol interactions in milk–coffee systems, supported mainly by spectroscopic and thermodynamic data. Could the authors elaborate on the limitations of current in vitro findings and suggest how future in vivo or omics-based validation might substantiate the proposed molecular mechanisms influencing bioaccessibility and antioxidant stability?

Answer:We appreciate the reviewers' suggestions. Sections 341-349 of the revised manuscript further elaborate on the limitations of existing in vitro experiments. Additionally, the Future Perspectives section 533-537 proposes future validation through in vivo experiments or omics studies to elucidate the molecular mechanisms influencing bioavailability and antioxidant stability. In the revised manuscript, these additions are highlighted in (RED)color.

Because much of the current evidence is derived from standardized in vitro digestion and spectroscopic/thermodynamic readouts that simplify real matrices and do not capture epithelial transport, active efflux, mucus dynamics, or phase-II metabolism, these practical levers should be optimized with caution. Likewise, benefits demonstrated under controlled in vitro conditions require corroboration in matched formulations to establish whether they translate to higher systemic exposure. Accordingly, direct in vivo or ex vivo transport confirmation is still needed to determine the extent to which gains in bioaccessibility are accompanied by genuine increases in bioavailability.Specific thresholds should be optimized case-by-case rather than assumed universally.

Additionally, future work should validate these mechanisms in milk–coffee matrices matched to real formulations and processing histories by combining in vivo and ex vivo transport/kinetic approaches (e.g., epithelial transport models and plasma metabolite profiling) with multi-omics integration (metabolomics, proteomics, lipidomics), thereby establishing quantitative in vitro–in vivo correlations and mechanistic attribution [100].

[100]Zhou, H., Tan, Y., & McClements, D. J. (2023). Applications of the INFOGEST in vitro digestion model to foods: A review. Annual Review of Food Science and Technology, 14(1), 135-156.

  • The manuscript outlines innovative non-thermal technologies (HPH, PEF, CP, ultrasound) and BAS for flavor and stability optimization. However, there is limited comparison of industrial scalability, cost-efficiency, and safety regulation compliance across these technologies. Could the authors include a concise comparative assessment table or commentary linking these processes to practical application potential?

Answer: We sincerely appreciate this valuable insight. We conducted a comparative analysis of existing processing technologies at industrial scale, focusing on cost-effectiveness, product impact, and compliance with safety regulations, and compiled the findings in Table 3: Comparative Assessment of Processing Routes for Milk-Coffee Beverages. In the revised manuscript, these additions are highlighted in (RED)color, specifically: Red fonts were added in line 424-435:

Across processes summarized in Supplementary Table S3, we compare industrial readiness, throughput/cost–energy considerations, and regulatory/validation requirements for UHT, BAS, HPH, PEF, cold plasma, and power ultrasound to link technology choice with practical deployment. Briefly, UHT remains the high-TRL baseline with clear global frameworks but the highest thermal load; BAS preserves aroma via aseptic post-mixing yet requires stricter hygienic design and media-fill validation; HPH is broadly line-ready and cost-efficient for droplet refinement and stability; PEF shows growing beverage-scale adoption with competitive energy at target lethality but needs product-specific process validation; cold plasma is pilot-to-early industrial, achieving near-ambient inactivation while demanding tight control of discharge parameters and packaging compatibility; and power ultrasound provides dispersion/microstructure benefits with economics governed by power density and residence-time design.

  • Deeth, H. C., & Lewis, M. J. (2017). High temperature processing of milk and milk products. John Wiley & Sons.
  • Burton, H. (2012). Ultra-high-temperature processing of milk and milk products. Springer Science & Business Media.
  • Pereda, J., Ferragut, V., Quevedo, J. M., Guamis, B., & Trujillo, A. J. (2007). Effects of ultra-high pressure homogenization on microbial and physicochemical shelf life of milk. Journal of dairy science, 90(3), 1081-1093.
  • Buckow, R., Chandry, P. S., Ng, S. Y., McAuley, C. M., & Swanson, B. G. (2014). Opportunities and challenges in pulsed electric field processing of dairy products. International Dairy Journal, 34(2), 199-212.
  • Misra, N. N., Patil, S., Moiseev, T., Bourke, P., Mosnier, J. P., Keener, K. M., & Cullen, P. J. (2014). In-package atmospheric pressure cold plasma treatment of strawberries. Journal of food engineering, 125, 131-138.
  • Chemat, F., Rombaut, N., Sicaire, A. G., Meullemiestre, A., Fabiano-Tixier, A. S., & Abert-Vian, M. (2017). Ultrasound assisted extraction of food and natural products. Mechanisms, techniques, combinations, protocols and applications. A review. Ultrasonics sonochemistry, 34, 540-560.

  • The review highlights emerging plant-based milk coffees but provides limited discussion on the compatibility and phase stability of plant proteins with coffee polyphenols. How might enzymatic modification or fermentation pre-treatments improve emulsification and flavor retention in these systems?

Answer: We We thank the reviewers for their valuable suggestions. In Section 4.4, we have added a discussion on the compatibility and co-stability of plant proteins with coffee polyphenols, as well as how enzyme modification or fermentation pretreatment can improve emulsification and flavor retention in these systems. In the revised manuscript, these additions are highlighted in (RED)color; specifically: Red fonts were added in line 492-503:

Plant-based milk coffee products, including those based on oat, soy, almond, or pea, are gaining attention due to environmental and health considerations. However, they frequently suffer from phase instability and flavor imbalance when mixed with acidic coffee. Plant-based milk coffee undergoes phase separation. Soy milk in coffee often curdles due to factors like acidity, temperature, concentration, and mixing order. This incompatibility largely arises from electrostatic aggregation and polyphenol–protein complexation. Enzymatic modification and lactic fermentation can effectively alleviate these issues by increasing protein solubility, modulating surface charge, and generating smaller peptides or exopolysaccharides that improve phase stability. These treatments also enhance flavor retention by reducing excessive binding with phenolics and preserving volatile compounds during heating or storage [57,71,107]. Brown et al. [65] found this phase separation reversible, as cooling or adjusting soy milk concentration can restore a uniform mixture.

  • The “Future Perspectives”section mentions integrating digital twins, LCA/TEA, and multi-omics analyses. Could the authors more clearly define how these tools can be operationalized within dairy and plant-based coffee production pipelines to support sustainability metrics and product traceability?

AnswerWe appreciate the reviewers' valuable questions. We agree that it is crucial to articulate the operational pathways of these tools more clearly. In the production of dairy-based and plant-based coffee:1.Multi-omics analysis employs proteomics, metabolomics, and other technologies to elucidate the key interaction mechanisms of biological macromolecules such as protein-polyphenols at the molecular level. This guides the enhancement of formulation functionality and product stability, while establishing unique biological "fingerprints" for raw materials to support product traceability and quality consistency control.2. Digital twins can integrate production line sensor data to construct real-time virtual models for dynamically monitoring and optimizing process parameters (such as sterilization temperature and homogenization pressure).  By incorporating blockchain technology, they enable end-to-end data traceability from raw materials to finished products, ensuring consistent quality while maximizing energy efficiency.3. LCA/TEA establishes a comprehensive and standardized evaluation framework to systematically quantify the environmental impacts (such as carbon and water footprints) across the entire life cycle of different formulations and process routes (e.g., UHT versus BAS comparison), as well as economic costs, thereby providing data support for sustainable and economically viable technology selection.These tools collectively form a synergistic framework spanning from molecular mechanisms to production systems, and further to comprehensive environmental-economic assessments, providing closed-loop guidance and decision support for developing the next generation of sustainable coffee beverages.Based on the above discussion, we will include the following content in the future outlook section, In the revised manuscript, these additions are highlighted in (RED)color; specifically: Red fonts were added in line 527-532:

Digital twins optimize processes and energy efficiency through real-time simulation, while integrating blockchain for full-chain traceability; LCA/TEA quantifies the environmental and economic costs of different processes to support sustainable decision-making; Multi-omics analyzes component interactions at the molecular level, guiding formulation design and establishing traceability fingerprints.  These three elements synergistically form a closed-loop guidance framework from mechanism to production.
